# Pattern and predictor of hookworm re-infection among schoolchildren in three districts of Amhara Region, northwest Ethiopia

**Shegaw Belay[1], Getaneh Alemu[2], Tadesse Hailu[2]** *

1 Amhara National Regional State Health Bureau, Bahir Dar Health Science College, Bahir Dar, Ethiopia,
2 Departments of Medical Laboratory Science, College of Medicine and Health Sciences, Bahir Dar University, Bahir Dar, Ethiopia

* tadessehailu89@gmail.com

## Abstract

### Background

Despite integrated prevention and control measures, the prevalence of hookworm is still high in Ethiopia. The re-infection rates and predictors are poorly addressed. Therefore, this study aimed to determine the patterns of hookworm re-infection rates and predictors among schoolchildren in northwest Ethiopia.

### Methods

A prospective follow-up study was conducted among 86 schoolchildren from May to December 2022. Data on predictors was collected using a questionnaire. Stool samples were collected and processed via Kato-Katz, McMaster, and spontaneous tube sedimentation techniques. All hookworm-infected children were treated with albendazole and followed for six months. The re-infection rates of hookworm were checked in the 4th and 6th months. Data was entered into Epi-data version 3.1 and analysed using SPSS version 25. Descriptive statistics were used to compute the re-infection rate. The associations of predictors with hookworm re-infection rates were calculated by logistic regression. Variables with a *p-value* < 0.05 were considered statistically significant.

### Results

Of the 86, 81 schoolchildren completed the study. The prevalence of hookworm re-infection in the 4th and 6th months was 23.5% and 33.3%, respectively. Living with family members greater than five (p = .017), poor utilization of latrine (p = .008), infrequent shoe wear (p = .039), and participating in irrigation (p = .020) were the predictors significantly associated with hookworm re-infections.

**Data Availability Statement:** All relevant data are within the paper and its Supporting Information files.

**Funding:** The author(s) received no specific funding for this work.

**Competing interests:** The authors have declared that no competing interests exist.

**Abbreviations:** AM, Artimetic mean; DALY, Disability-Adjusted Life-Years; KK, Kato-Katz; MM, McMaster; PSAC, preschool-age children; SAC, school-age children; STH, soil-transmitted helminth; STST, spontaneous tube sedimentation technique; WHO, World Health organization.

## Conclusions

The re-infection rate was high during the fourth and sixth months. Participating in irrigation, infrequent shoe wear, and poor latrine utilization were predictors of hookworm re-infection. Therefore, mass drug administration, regular shoe wearing, and health education should be advocated.

## Introduction

Hookworm is one of the soil-transmitted helminths (STHs) that dwell in the small intestine of human beings. The two main spp. of hookworm that infect humans are *Ancylostoma duodenale* and *Necator americanus*. They mainly affect the poorest and most deprived communities with poor nutrition, inadequate sanitation, overcrowding, and being barefoot [1–3].

Hookworm species were responsible for more than 845,000 disability-adjusted life years (DALYs) annually and were estimated to infect 229 million people globally [4]. Hookworm infection affects almost 120 million people in sub-Saharan Africa [5].

Ethiopia had the third highest sub-Saharan African hookworm infection burden [6]. Despite control measures including mass drug administration, water, sanitation, and hygiene community based health education implemented in the country, the prevalence of hookworm among children remains high. The problem is severe in poor and rural communities [7, 8]. Previous, reports in Ethiopia indicated that the prevalence of hookworm was high. For instance, the prevalence of hookworm spp. was 15.3% in southern Ethiopia, 48.2% in west Gojjam, 46.9% in Durbete town, 41.3% in Sebatamit, and 64.2–87.7% in rural Bahir Dar, Amhara Region [8–11].

Hookworm re-infection rates reveal continuous exposure to the sources of infections in endemic communities [12, 13]. Substantial re-infection after complete deworming of hookworm infection is a common problem [14, 15]. The re-infection can occur as early as two months after treatment in areas with high transmission settings [16]. For example, a study conducted in Malaysia showed that 51.8% of the treated population were re-infected by hookworm in the sixth month, after deworming. Similarly, 5.1% of schoolchildren were re-infected by hookworm 6 months post-treatment in Yunnan, People's Republic of China [16].

Several factors, including socio-demographic, socioeconomic, behavioral, and environmental factors, could be predictors of hookworm re-infection [17]. Hence, the prevention of hookworm requires integrated prevention, which includes chemotherapy, implementing improved water, sanitation, and hygiene (WASH), and a community-based health education strategy in endemic areas [18]. Although our study area is endemic for hookworm infection, the re-infection rates of hookworm and associated factors contributing to the re-infection are not yet well determined. Therefore, this study aimed to determine the re-infection rate and predictors of hookworm re-infection among schoolchildren in three endemic districts of Amhara Region, northwest Ethiopia.

## Materials and methods

### Study design, period, and area

A school-based prospective follow-up study was conducted from May to December 2022 to determine the re-infection rates of hookworm and associated factors in three selected districts of Amhara Region, Northwest Ethiopia. The altitude of Bahir Dar city administration is 1820

m above sea level, with 1839 mm mean annual rainfall and a temperature range of 14–28.1˚C. North Mecha district is also located 35 km southwest of Bahir Dar city with 1,800–2,500 m altitude, 1850 mm of rainfall, and a temperature of 24˚C. Bahir Dar Zuria is located at 1900–2700 m, with 1035 mm of rainfall and a temperature of 23˚C.

The sample size was calculated using 41.3% hookworm prevalence from a previous study [19], 95% level of confidence, a 5% margin of error, and a 15% non-respondent rate. Among 432, 403 schoolchildren were volunteered and examined to check for hookworm infection. All hookworm positive schoolchildren in selected primary schools, whose ages ranged from 6 to 14 years, lived for the last six months in the study area prior to data collection, and whose parents/guardians gave written consent for their children to participate and take albendazole were included in the study. Lastly, schoolchildren who gave assent and a stool sample in the fourth and sixth months post treatment were included. Those schoolchildren who received any anti-helminthic drug within the last three months prior to data collection were excluded from the study.

## Sample size and sampling technique

Three districts, namely: North Mecha, Bahir Dar City administration, and Bahir Dar zuria districts, were selected as study sites. One school from each district was randomly selected by a lottery method. The sample size was proportionally allocated to each school by considering the total number of students. A systematic random sampling technique was used to select participants in each school by using the class roster as a sampling frame. A stool sample was collected from each schoolchild and those children positive for hookworm infection were treated with albendazole. A total of 86 albendazole treated and cured from hookworm infection schoolchildren who provide assent to participate in the follow-up study were included (Fig 1).

## Data collection methods

A structured questionnaire was used to collect data on socio-demographics and factors associated with hookworm re-infection. The data was collected by trained laboratory professionals. Fresh stool samples were collected using a clean and leak-proof stool container labelled with the participant's unique identification number for each study participant. Stool samples were collected and transported to the laboratory center within an hour of collection. Each stool sample was processed and examined following standard operating procedures. Three sensitive diagnostic methods were used for the purpose of diagnosis.

**Spontaneous tube sedimentation technique (STST).** Nearly three grams of fresh stool sample were weighed, homogenized in 10 ml of normal saline solution, and mixed well. The mixture was filtered through a wire mesh into a 50 ml falcon tube, which was then filled with more saline solution up to a 50 ml gauge, plugged, and shaken vigorously. The falcon tube was left to stand for 45 minutes to settle. The sediment was taken with a Pasteur pipette, put on a microscope slide, and examined with 10 x followed by 40 x objectives of a microscope to check for the presence of ova of hookworm [20].

**Kato-Katz (KK).** About two-three grams of fresh stool sample were pressed through a mesh screen to remove large particles. Around 41.7 mg of stool was sieved and transferred to the template, which was put on a slide until the template hole was filled. Then, the template was removed and the stool sample was covered and pressed with cellophane, which was previously immersed overnight in glycerol-malachite green. The KK smears were examined within 30–60 minutes for hookworm spp. [21].

**McMaster (MM).** Approximately, two grams of fresh stool were suspended in 30 ml of saturated salt (NaCl) solution at room temperature. The suspension was filtered through a

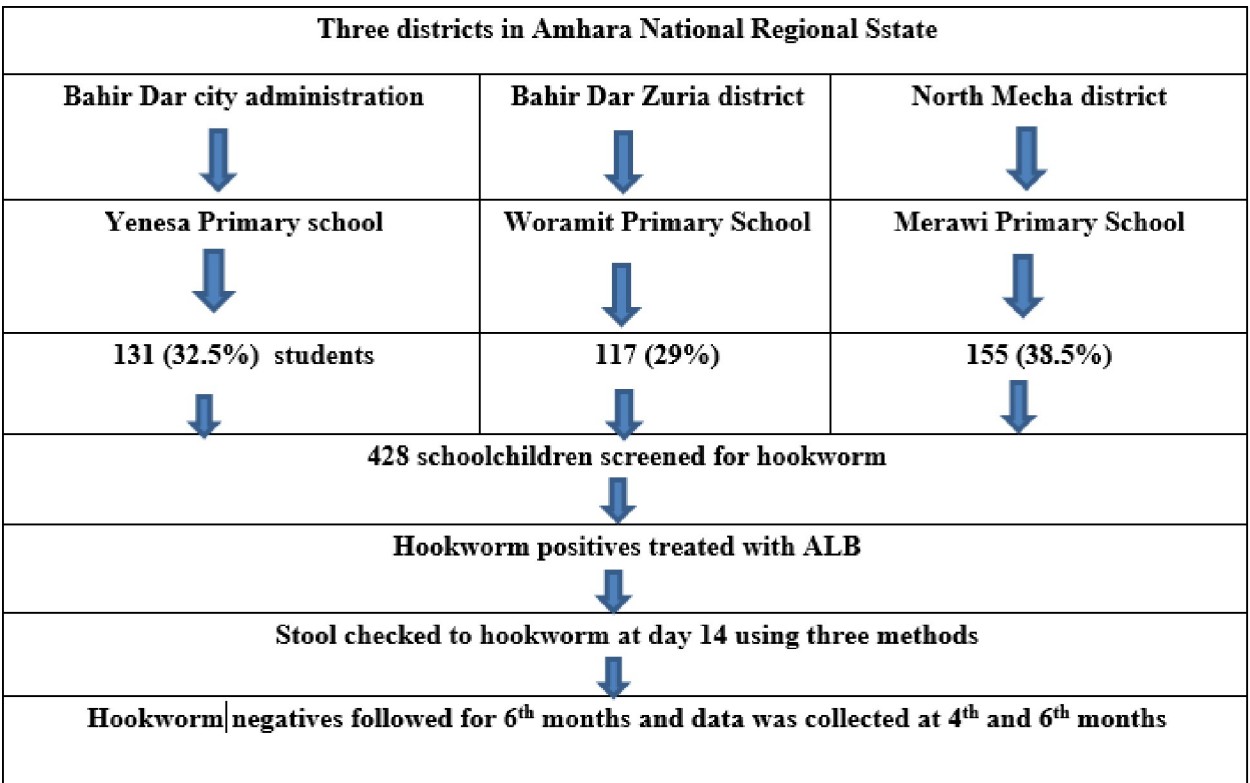

**Fig 1. Sampling frame that shows how study sites, schools and schoolchildren were selected, 2022.**

mesh and well mixed 10 times using a Pasteur pipette. A 0.15 ml aliquot was added to each side of a MM slide chamber, waited two minutes to settle, and was examined under a light microscope with 10x objectives. The faecal egg count of hookworm spp. was calculated by multiplying the faecal egg count by 50 [22].

## Re-infection rate assessment

Children that had positive hookworm infections during the baseline survey were treated with albendazole and microscopic negatives (cures) after 14 days of post-treatment were re-tested in the 4th and 6th months follow-up surveys using the same laboratory techniques.

$$\text{Re–infection rate} = \frac{\text{Number of infected children after treatment}}{\text{Number of negative children at follow–up who}} \text{X100}$$
$$\text{were positive at baseline}$$

## Data quality control

Training was given for both questionnaire-based and laboratory-based data collectors. The questionnaire data was checked daily for completeness by the principal investigator. The amount of stool sample and consistency were checked. Two laboratory personnel examined the slides independently. The result of their observation was recorded separately for later comparison. All discordant results were re-checked. Ten percent of the slides were cross-checked by the principal investigator. Generally, the quality and reliability of the finding were checked at pre-analytical, analytical, and post-analytical steps.

## Data management and analysis

All data were entered into Epi Info version 3.1 and analyzed with SPSS version 25 software. Descriptive statistics were used to compute the re-infection rate. The associations of independent variables with the re-infection rate of hookworm were calculated by bivariate logistic regression analysis. All variables with *P-values* < 0.20 in the bivariate analysis were run in a multivariate logistic regression analysis to filter the confounding effects. Variables with a *p-value* < 0.05 were considered statistically significant.

## Ethical consideration

Ethical clearance was obtained from the College of Medicine and Health Sciences Institutional Review Board, Bahir Dar University (protocol number: 384/2022). Permission letters were also obtained from the Bahir Dar City administration, North Mecha district, and Bahir Dar Zuria district health and education offices. Furthermore, written informed consent from the parent/guardian and assent from schoolchildren were secured. Confidentiality of the collected information and laboratory test results were maintained. Hookworm positive cases were treated with a single-dose albendazole (400 mg). Finally, those who tested positive for any intestinal parasitic infefctions were referred to nearby healthcare facilities for treatment.

## Results

### Socio-demographic characteristics of study participants

Among the 423 schoolchildren in this study, 403 provided complete data and provided stool samples in the baseline survey, with a response rate of 94.2%. Among the 403 participants, 223 (55.3%) were males. The mean age of participants was 10.84 ± 2.04 SD years. The study participants were selected from three districts in Amhara Region: 131 (32.5%) participants from Bahir Dar city administration, 155 (38.5%) participants from North Mecha district, and 117 (29%) participants from Bahir Dar Zuria district (Table 1).

The prevalence of hookworm infection among schoolchildren from the baseline data was 115 (27.2%). A total of 86 (33 female and 53 male) participants who were hookworm positive at baseline, treated with albendazole and tested negative after 14 days of post-treatment were followed for 6 months to determine their re-infection rate status. Stool sample was checked in the fourth and sixth months. Five students were lost from follow-up in the fourth month data collection period. Eighty-one (28 female and 53 male) participants gave a stool sample and completed the hookworm re-infection rate follow-up (Fig 2). The majority of participants

**Table 1. Socio-demographic characteristics of schoolchildren participated in the baseline survey attending selected schools in Amhara Region, northwest Ethiopia, from May to December 2022 (N = 403).**

| Variables | | Frequency n (%) | IP n (%) |
|---|---|---|---|
| Sex | Male | 223 (55.3) | 104 (46.6) |
| | Female | 180 (44.7) | 64 (35.6) |
| Age | 6–10 | 156 (38.7) | 45 (28.8) |
| | 11–14 | 247 (61.3) | 123 (49.8) |
| Residence | Rural | 308 (76.4) | 134 (43.5) |
| | Urban | 95 (23.6) | 34 (35.8) |
| Participant districts | North Mecha | 155 (38.5) | 53 (34.2) |
| | Bahir Dar city | 131 (32.5) | 51 (38.9) |
| | Bahir Dar Zuria | 117 (29.0) | 64 (54.7) |

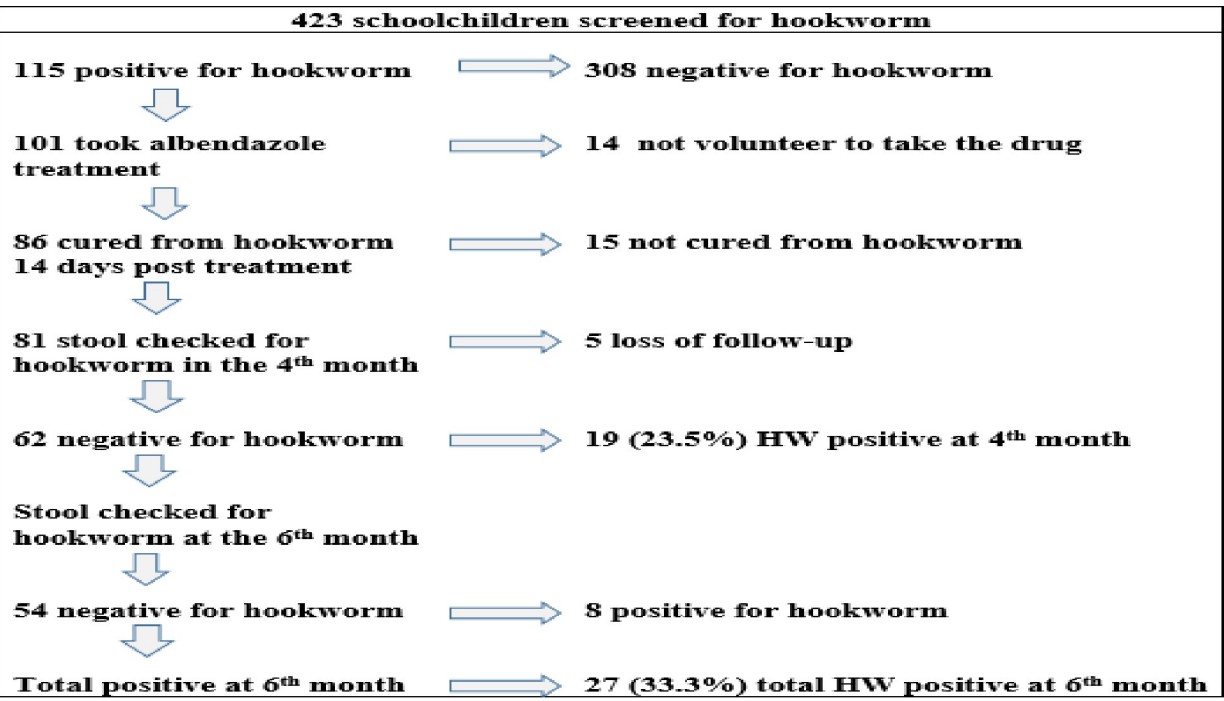

**Fig 2. Sampling frame of study participants in the hookworm re-infection study, 2022.**

were males (65.4%), 11–14 age group (82.7%), rural dwellers (75.3%) and North Mecha district (38.3%).

## Re-infection rate and intensity of hookworm species

The overall re-infection rate of hookworm species among schoolchildren was 23.5% (95% CI 14.8–34.2) in the 4th month and 33.3% (95% CI 23.2–44.7) in the 6th month by the combination of three methods. The re-infection rate of hookworm was 21% by KK, 23.5% by STST, and 23.5% by MM in the 4th month. In the 6th month, the re-infection rate was 29.6% by KK, 33.3% by STST, and 33.3% by MM (Figs 2 and 3). In the six-month follow-up, 3 (3.7%) *S. mansoni*, 3 (3.7%) *A. lumbricoides*, and 1 (1.2%) *T. trichuria* were also detected.

## The intensity of infection

The arithmetic mean intensity of hookworm re-infections in the 4th month was 53.7 EPG and 139.9 EPG by the KK and MM methods, respectively. In the 6th month, the arithmetic mean intensity of infection was increased to 92.6 EPG and 207.1 EPG by the KK and MM methods, respectively. The arithmetic mean intensity of hookworm re-infections was grouped under "light infection intensity" in the 4th and 6th month follow-up periods based on WHO thresholds for determining infection intensity (Fig 4).

The re-infection rates of hookworm were 19.4% (95% CI 7.5–37.5), 20.8% (95% CI 7.1–42.2), and 30.8% (95% CI 14.3–51.8) at North Mecha district, Bahir Dar city administration, and Bahir Dar Zuria district, respectively, in the 4th month. In the 6th month, the re-infection rate was 29% (95% CI 14.2–48.0) in the North Mecha district, 42.3% (95% CI 23.4–63.1) in the Bahir Dar zuria district, and 29.2% (95% CI 12.6–51.1) in the Bahir Dar city administration (Fig 5).

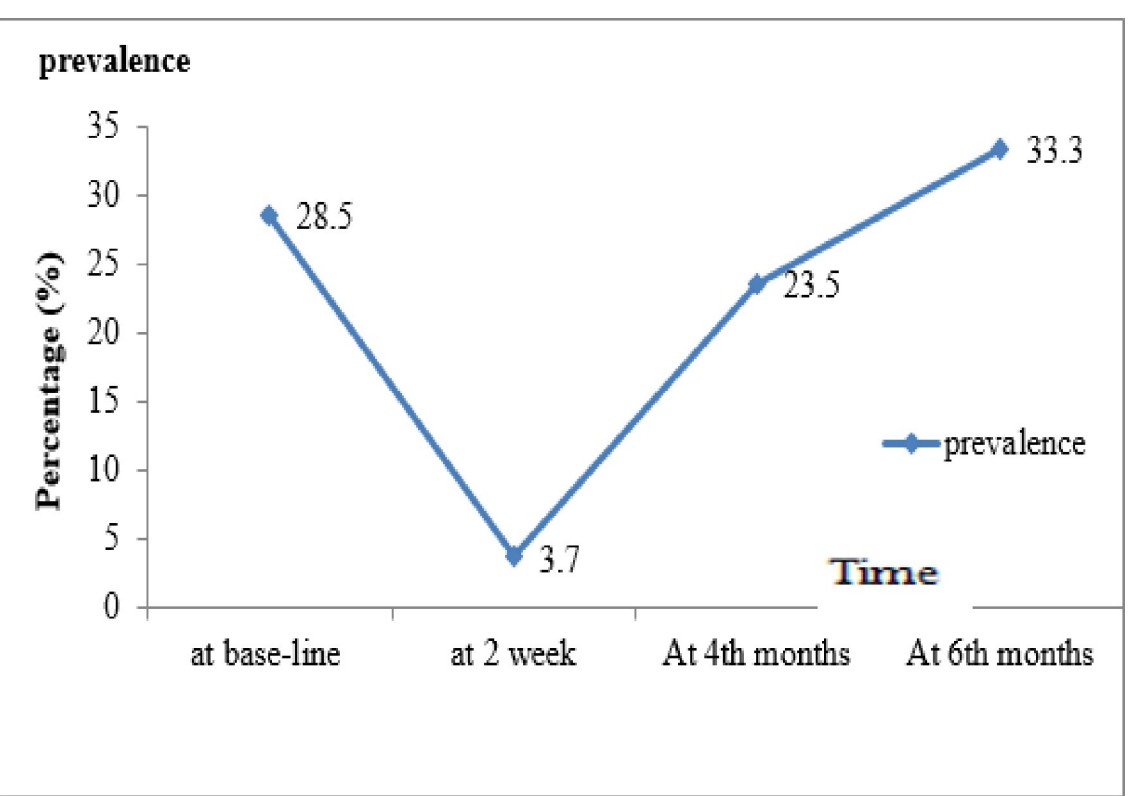

**Fig 3. Prevalence of hookworm at baseline, 2 weeks, and 4th and 6th months after deworming by composite reference method.**

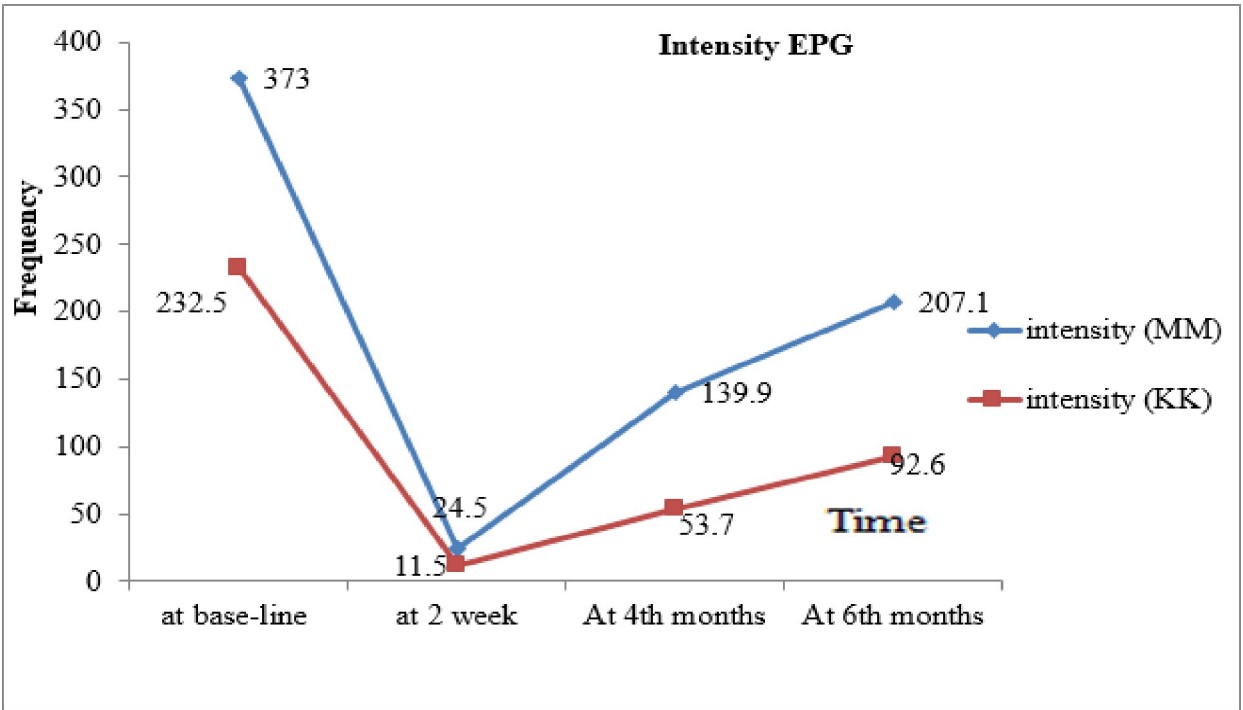

**Fig 4. Arithmetic mean intensity of hookworm (EPG) at baseline, two weeks post treatment, 4th, and 6th months after deworming by KK and MM diagnostic method, 2022.**

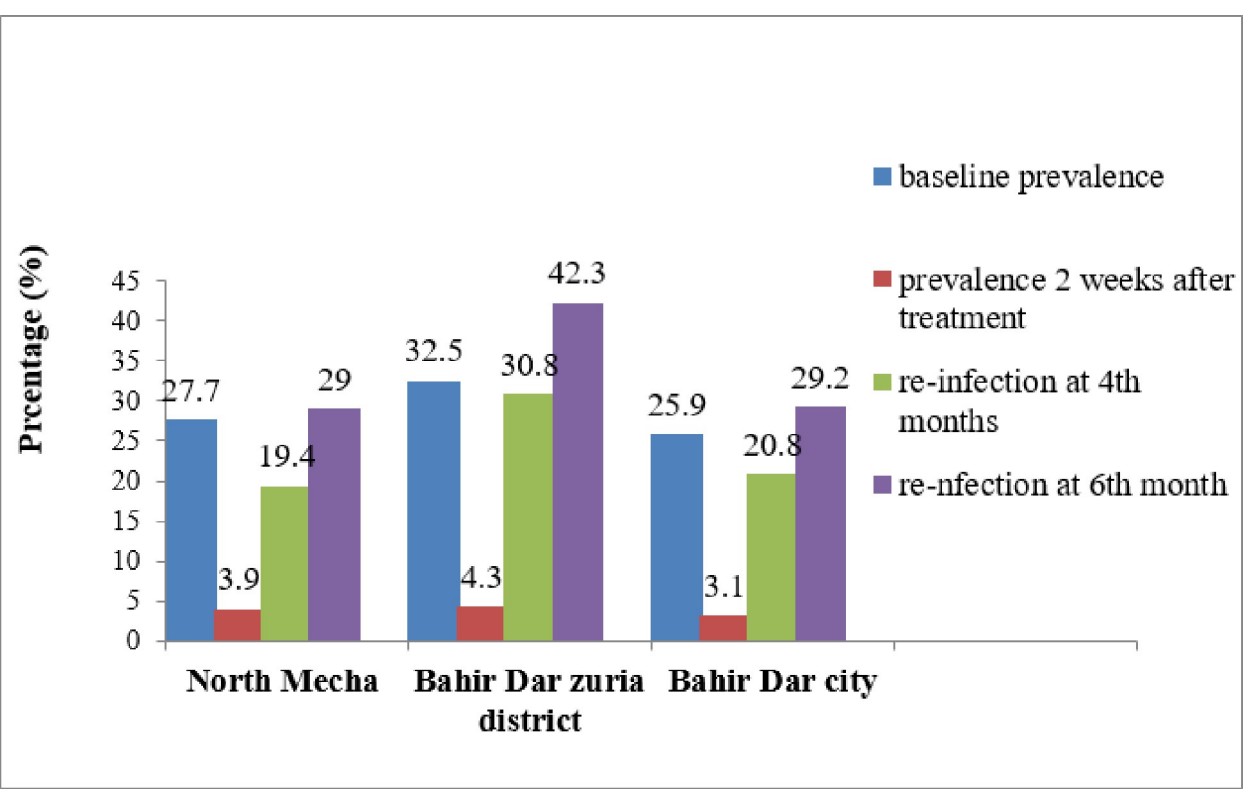

**Fig 5. Baseline prevalence, two weeks after treatment, in the 4th, and 6<sup>th</sup> month re-infection rates of hookworm in Amhara Region, northwest Ethiopia.**

### Factors associated with hookworm re-infection

In the bivariate analyses, schoolchildren who practice open defecation (COR = 3.186 CI = 1.023–9.927, p = .046), poor hand washing practice after using a toilet (COR = 2.941 CI = 1.020–8.484, p = .046), infrequent shoe wearing (COR = 3.877 CI = 1.241–12.109), p = .02), participate in irrigation (COR = 3.400 CI = 1.091–10.60, p = .035) had higher re-infection rates of hookworm infection than their counterpart in the 4<sup>th</sup> month.

In the multivariate analysis, children who did not wear shoes consistently were 3.4 times more likely to be re-infected by hookworm (AOR = 3.359 CI = 1.024–11.026, p = .020) in the 4<sup>th</sup> month. Similarly, schoolchildren who participate in irrigation activity were 3.7 times more likely to be re-infected by hookworm (AOR = 3.704 CI = 1.092–12.562, p = .036) than schoolchildren who did not participate in irrigation in the 4<sup>th</sup> months (Table 2).

In the multivariate analysis, schoolchildren who live within more than five family members (AOR = 4.682 95% CI = 1.322–16.578, P = .017), have poor latrine utilization (AOR = 5.088 95%CI = 1.517–17.065, P = .008), do not wear shoes consistently (AOR = 3.099 95% CI = 1.058–9.083, P = .039), and actively participate in irrigation activities (AOR = 4.142 95% CI = 1.247–13.753, P = .020) were significantly associated with hookworm re-infection rates than their counterparts at the sixth month post treatment (Table 3).

## Discussion

Hookworm infection is a treatable and preventable disease with high public health importance in endemic countries like Ethiopia. The national control program was designed to eliminate

**Table 2. Bivariate and multivariate analysis of factors associated with hookworm re-infection in the 4th month among schoolchildren in the Amhara Region, northwest Ethiopia, 2022 (n = 81).**

| Variables | Re-infection rates of hookworm species in 4th month | | | | | | |
|---|---|---|---|---|---|---|---|
| | Categories | Pos. | Neg. | COR (95% CI) | *p*-value | AOR (95% CI) | *p*. value |
| | | n (%) | n (%) | | | | |
| Sex | Female | 5 (17.9) | 23 (82.1) | 1 | | | |
| | Male | 14 (26.4) | 39 (73.6) | 1.651 (.397–3.574) | .754 | | |
| Age | 6–10 | 3 (21.4) | 11 (78.6) | 1 | | | |
| | 11–14 | 16 (23.9) | 51 (76.1) | 1.150 (.285–4.640) | .844 | | |
| Family size | = <5 | 5 (14.7) | 29 (85.3) | 1 | | | |
| | >5 | 14 (19.8) | 33 (70.2) | 2.461(.790–7.667) | .120 | 3.010(.874–10.369) | .081 |
| Residence | Urban | 2 (10) | 18 (90) | 1 | | 1 | |
| | Rural | 17 (27.9) | 44 (72.1) | 3.477(.727–16.621) | .118 | 1.603(.286–8.978) | .591 |
| Latrine utilization | Yes | 5 (11)2 | 33 (88.8) | 1 | | | |
| | No | 14 (32.6) | 29 (67.4) | 3.186 (1.023–9.927) | .046 | 2.746 (.789–9.555) | .112 |
| Hand washing after the toilet | Yes | 9 (16.7) | 45 (83.3) | 1 | | 1 | |
| | No | 10 (37) | 17 (63) | 2.941 (1.020–8.484) | .046 | 1.585 (.472–5.326) | .456 |
| Participation in irrigation | No | 5 (17.3) | 34 (82.7) | 1 | | 1 | |
| | Yes | 14 (33.3) | 28 (66.7) | 3.400 (1.091–10.60) | .035 | 3.704 (1.092–12.56) | .036* |
| Frequency of shoe wearing | Always | 5 (12.2) | 36 (87.8) | 1 | | 1 | |
| | Sometimes | 14 (35) | 26 (65) | 3.877(1.241–12.109) | .020 | 3.359 (1.024–11.026) | .046* |

* Pos = positive, Neg = negative, CI = confidence interval, COR = crude odds ratio, AOR = adjusted odds ratio, * = significant association

this parasitic disease by 2025 and achieve less than 2% prevalence using a single dose of ALB (400 mg) or MEB (500 mg), WASH, health education, and advocacy [23]. Reports showed that re-infections of hookworm occur quickly after treatment and significantly affect the success of preventive chemotherapy [15, 16].

**Table 3. Bivariate and multivariate analysis of factors associated with hookworm re-infection in the 6th month among schoolchildren in the Amhara Region, northwest Ethiopia (n = 81).**

| Variable | Re-infection rates of hookworm species in 6th month | | | | | | |
|---|---|---|---|---|---|---|---|
| | Categories | Pos. n (%) | Neg. n (%) | COR (95%CI) | *p*-value | AOR 95% CI | *p*. value |
| Family size | <5 | 7 (20.6) | 27 (79.4) | 1 | | 1 | |
| | >5 | 20 (42.6) | 27 (57.4) | 2.857 (1.038–7.865) | .042 | 4.682 (1.322–16.578) | .017* |
| Residence | Urban | 3 (15) | 17 (85) | 1 | | 1 | |
| | Rural | 24 (39.3) | 37 (60.7) | 3.676 (.972–13.905) | .055 | 1.535 (.328–7.178) | .586 |
| Latrine availability | Yes | 6 (16.2) | 31 (83.8) | 1 | | 1 | |
| | No | 21 (47.7) | 23 (52.3) | 4.717 (1.642–13.55) | .004 | 1.402 (.263–7.484) | .692 |
| Latrine utilization | Yes | 6 (15.8) | 32 (84.2) | 1 | | 1 | |
| | No | 21 (48.8) | 22 (51.2) | 5.091(1.769–14.65) | .003 | 5.088 (1.517–17.065) | .008* |
| Hand wash after toilet | Yes | 15 (27.8) | 39 (72.2) | 1 | | 1 | |
| | No | 12 (44.4) | 15 (55.6) | 2.080 (.793–5.458) | .137 | .770 (231–2.564) | .671 |
| Participation in irrigation | No | 8 (20.5) | 31 (79.5) | 1 | | 1 | |
| | Yes | 19 (45.2) | 23 (54.8) | 3.201 (1.194–8.585) | .021 | 4.142 (1.247–13.753) | .020* |
| Frequency of shoe wearing | Always | 8 (19.5) | 33 (80.5) | 1 | | 1 | |
| | Sometimes | 19 (47.5) | 21 (52.5) | 3.732 (1.386–10.052) | .009 | 3.099 (1.058–9.083) | .039* |

COR = crude odds ratio, AOR = adjusted odds ratio

* = significant association

The baseline prevalence of hookworm infection among schoolchildren was 27.2% (95% CI 23.0–31.7) by the composite reference method. This result is in line with the study conducted in South Sumatra, Indonesia (25.9%) [24] and North Gondar (26.2%) [11]. However, the current result is lower than the studies in Debre Elias and Sanja districts, Amhara Region (63.2%) [25], and rural communities in Bahir Dar, Amhara Region (46.9%) [10]. The variation could be attributed to differences in study participant age (the previous study included adults), parasite endemicity (level of soil contamination), diagnostic methods used, and health status of participants (Zeleke used five methods, and the study participants were symptomatic adults). On the contrary, the current finding is higher than the study in Central Kenya (1%) [3] and in Bibugn, Amhara Region (7.5%) [26]. This variation could be attributed to the diagnostic method (only the KK method was used in Kenya, but three methods were used in the current study), temperature, and altitude (in Bibugn, there is cold weather at 9–24˚C, which is not comfortable for hookworm larvae survival, development, and transmission).

The post-treatment re-infection patterns of hookworm species largely depend on the degree of endemicity within the community and the level of soil contamination. In the current endemic districts of the Amhara region, follow-ups in the 4th and 6th months after the administration of single-dose 400 mg ALB revealed that re-infection with hookworm was rapid in terms of prevalence. However, the intensity was light. Indeed, the re-infection rate of hookworm reached 23.5% and 33.3% in the 4th and 6th months after treatment, respectively. This increase in hookworm re-infection in the 4th and 6th months of follow-up was probably attributable to the favorable climatic conditions, convenient soil temperatures and humidity in the summer season (June–October), the soil type in the study area (clay and sandy), and the level of soil contamination in the area. This condition favors the survival, development, and transmission of hookworm larvae. Furthermore, the school was closed in the summer season, and children spent most of their time in the field (they defecate in an open field at a farm, which is conducive to parasite survival and distribution). Moreover, there is also a habit of walking barefoot in the agricultural area in the summer; this could be the cause of the high hookworm re-infection rate during the study period.

The re-infection rate of hookworm in the 4th month among the schoolchildren was 23.5% (95% CI 14.8–34.2%), which is consistent with previous studies in Tanzania (25.0%) [27] and Debre Elias District, Amhara Region (21.4%) [28]. However, this result is higher than that of the Chinese study (2%) (16). The difference might be due to seasonal differences. In China, the study was conducted from December–March, which is a cold and dry season that is unsuitable for the development and survival of hookworm larvae), the baseline prevalence, the level of soil contamination, and the dose of ALB (the previous study used a triple dose). This result is also higher than the study in Chencha district, southern Ethiopia (1%) [29]. The difference might be due to short follow-up time (only at the 3rd month), STH species differences, and diagnostic method differences. In Chencha, re-infection was assessed at the 3rd month and used only the KK method, but there is one extra month in this study that allows more time for schoolchildren to become newly re-infected and for surviving infections to re-establish and release eggs. Also, two diagnostic methods (KK and MM which increase the detection level) were used for the diagnosis method in the current study [30].

In the 6th month, the re-infection rate of hookworm was 33.3% (95% CI 23.2–44.7). This result is in line with the study in Myanmar (28.07%) [17]. However, this result was lower than the study in Malaysia (51.8%) [31], and a (55%) re-infection was obtained from a systematically reviewed and meta-analysed research report [32]. The difference could be due to study time variation, during which many intervention activities like mass drug administration (MDA), WASH, and community based health education have been done in the country. This decreases hookworm prevalence through implementing the MDA program among school-age children, creating an open defecation-free environment, and house-to-house education by

health extension workers. On the contrary, this result is higher than the study in Yunnan, China (5.1%) [16]. The difference might be due to the study season difference; the previous study was done in the winter season (December–March), which was a cold and dry condition known to be unsuitable for the development and survival of hookworm larvae. Similarly, the current finding is higher than the study in northwest Ethiopia, at 7.5% after one year [33]. The difference could be due to the diagnostic methods and species differences in Zeleke's study, in which direct wet mount and KK methods were used.

In both the 4th and 6th months, there was a slightly higher rate of re-infection in the Bahir Dar Zuria district than in the North Mecha district and Bahir Dar city administration in the current study. In the 4th month, Bahir Dar Zuria district participants were nearly 1.7 and 1.9 times more likely to be re-infected by hookworm than participants from Bahir Dar city administration and north Mecha district, respectively. At the same time, Bahir Dar Zuria district participants were 1.8 times more likely to be re-infected than Bahir Dar city administration and north Mecha district participants in the 6th month. But the difference was not statistically significant. A slight increase in Bahir Dar Zuria might be due to the initial prevalence and level of environmental sanitation in the school compound, where there was open defecation. During the visits to the study areas, we observed that the school's hygiene was poor; the class floor was covered in mud and dust, many of the students were barefooted, and there was no water for drinking or hand washing after using the toilet in the school compound. Besides, the distribution of hookworm among STHs is higher in the current study areas due to the altitude. A high prevalence of hookworm is reported in areas less than 2000 meters above sea level, especially in semi-highland areas [34].

The present study also assessed factors for hookworm re-infection and revealed that the frequency of shoe wearing had a significant negative association with hookworm re-infection. Study participants who do not always wear shoes were 3.4 times more likely to be re-infected by hookworm in the 4th month than those who always wear shoes. This finding agrees with study findings in Kenya [35], Shewa, Ethiopia [36], and western Ethiopia [37]. This is because wearing shoes minimizes hookworm transmission by providing a physical barrier and preventing larval penetration through the exposed skin of the feet, thus reducing the risk of infection. Similarly, participants who participated in farming were 3.7 times more likely to be re-infected by hookworm than those who did not participate in agriculture. This result agrees with the study in Timor-Leste [38]. The reason could be due to the open defecation behavior of people; most adults deposit their faces in the agricultural fields when performing their activity (an agricultural field is ideal for parasite survival and distribution).

In the 6th month, living with more than five family members and poor utilization of the latrine emerged as risk factors for hookworm re-infection. Study participants with a family size greater than five were 4.7 times more likely to be re-infected by hookworm than individuals with five family members or fewer. This finding is consistent with the studies in Kenya [35] and western Ethiopia [37]. The reason could be the difficulty of buying shoes for the entire family as the family size increases, and the children will be barefooted; this increases hookworm re-infection, and also, if one family member is infected, the transmission will be probable. Similarly, children who defecated in the open field were five times more likely to be re-infected with hookworm than those who used latrines. This finding agrees with the study conducted in Chencha, southern Ethiopia [29]. This is because latrine use minimizes hookworm transmission from open defecation, which is conducive to parasite distribution.

## Conclusions

The re-infection rate of hookworm was high during the 4th and 6th months after treatment. The intensity of infection during the 4th and 6th months' detection was light. Participating in

irrigation and irregular shoe wear were predictors of hookworm re-infection in the 4[th] month. In addition, participating in irrigation, irregular shoe wear, poor latrine utilization, and living with large family sizes were also risk factors for hookworm re-infection in the 6[th] month. Therefore, MDA, regular shoe-wearing and health education on behavioral change interventions are strongly recommended.

## Acknowledgments

We thank Bahir Dar University, College of Medicine and Health Sciences, School of Health Sciences, Department of Medical Laboratory Science for ethical approval of the study; study participants for volunteer participation in the study; Amhara National Regional Health Bureau for partial funding.

## Author Contributions

**Conceptualization:** Getaneh Alemu, Tadesse Hailu.

**Data curation:** Shegaw Belay, Tadesse Hailu.

**Formal analysis:** Shegaw Belay.

**Funding acquisition:** Shegaw Belay.

**Investigation:** Shegaw Belay.

**Methodology:** Shegaw Belay.

**Project administration:** Shegaw Belay.

**Resources:** Shegaw Belay.

**Software:** Getaneh Alemu.

**Supervision:** Getaneh Alemu, Tadesse Hailu.

**Validation:** Getaneh Alemu, Tadesse Hailu.

**Visualization:** Getaneh Alemu.

**Writing – original draft:** Shegaw Belay.

**Writing – review & editing:** Getaneh Alemu, Tadesse Hailu.

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
