## [Decision Letter · Decision Letter 0]

28 Feb 2024

PONE-D-23-41533Pattern and predictor of hookworm re-infection among schoolchildren in three districts of Amhara Region, northwest EthiopiaPLOS ONE

Dear Dr. Jember,

Thank you for submitting your manuscript to PLOS ONE. After careful consideration, we feel that it has merit but does not fully meet PLOS ONE’s publication criteria as it currently stands. Therefore, we invite you to submit a revised version of the manuscript that addresses the points raised during the review process.Try to address all comments raised by the reviewer, focusing on the introduction, method and discussion. Please submit your revised manuscript by Apr 13 2024 11:59PM. If you will need more time than this to complete your revisions, please reply to this message or contact the journal office at plosone@plos.org. Please include the following items when submitting your revised manuscript:A rebuttal letter that responds to each point raised by the academic editor and reviewer(s). You should upload this letter as a separate file labeled 'Response to Reviewers'.A marked-up copy of your manuscript that highlights changes made to the original version. You should upload this as a separate file labeled 'Revised Manuscript with Track Changes'.An unmarked version of your revised paper without tracked changes. You should upload this as a separate file labeled 'Manuscript'.

We look forward to receiving your revised manuscript.

Kind regards,

Tsegaye Alemeyhu, Msc

Academic Editor

PLOS ONE

Journal Requirements:

2. We note that another paper by the same author group and with the title [“Performance Evaluation of Three Diagnostic Methods for Soil-Transmitted Helminth Infections among Schoolchildren in Amhara Region, Northwest Ethiopia] was recently published. It has been noted that the two submissions may have some overlap with regard to the population cohort.

We ask that you please clarify why you feel the two reports should be considered as separate articles in your cover letter. In your manuscript, please ensure you cite, discuss, and acknowledge overlap with the related work, and provide adequate justification (in the Introduction) for the new submission in light of the related published work.

For more information about our policy on related manuscripts please see http://journals.plos.org/plosone/s/ethical-publishing-practice#loc-submission-and-publication-of-related-studies.

Additional Editor Comments:

Dear Dr Tadesse H,

Thank you for submitting your manuscript to PLOS ONE. Now the reviewer has evaluated the manuscript and suggested it be revised before deciding whether it is published or not. Therefore, address all the comments raised by the reviewers. A point-by point response for each comment should be submitted.

Reviewers' comments:

Reviewer's Responses to Questions

**Comments to the Author**

1. Is the manuscript technically sound, and do the data support the conclusions?

Reviewer #1: Partly

Reviewer #2: Yes

Reviewer #3: Yes

2. Has the statistical analysis been performed appropriately and rigorously? 

Reviewer #1: Yes

Reviewer #2: I Don't Know

Reviewer #3: I Don't Know

3. Have the authors made all data underlying the findings in their manuscript fully available?

Reviewer #1: No

Reviewer #2: Yes

Reviewer #3: Yes

4. Is the manuscript presented in an intelligible fashion and written in standard English?

Reviewer #1: Yes

Reviewer #2: Yes

Reviewer #3: No

5. Review Comments to the Author

Reviewer #1: The study by Belay et. al., titled "Patterns of Hookworm Reinfection Rates and Predictors among Schoolchildren in Northwest Ethiopia" aims to investigate the re-infection rates and predictors of hookworm among schoolchildren in northwest Ethiopia. While the study addresses an important public health issue, there are a few queries related to its design and methodology to assess the robustness of the findings and the implications for future research and interventions.

Major Queries:

External Validity: To what extent can the findings of the study be generalized to other populations of schoolchildren in Ethiopia or similar settings with high hookworm prevalence? What measures were taken to ensure the representativeness of the study sample?

Methodological Rigor: How were potential sources of bias, such as selection bias or measurement bias, addressed during the study design and data collection phases? Were steps taken to minimize the risk of misclassification or misreporting of predictor variables?

Longitudinal Dynamics: Given the short follow-up period of six months, how do the observed re-infection rates and predictors align with longer-term patterns of hookworm transmission and control? What implications do these findings have for the sustainability of intervention strategies over time?

Comprehensive Assessment: Are there additional predictor variables or potential confounders that were not included in the analysis but may significantly influence hookworm re-infection rates among schoolchildren? How could future research efforts broaden the scope of predictor assessment to capture a more comprehensive range of factors?

Intervention Strategies: Based on the identified predictors of hookworm re-infection, what targeted intervention strategies or public health interventions could be implemented to reduce transmission rates and improve outcomes among schoolchildren in northwest Ethiopia? How can these strategies be tailored to address the specific needs and challenges of the local community? Some discussion about them should be included in the manuscript.

Minor point : Line 124 -the word principal investigator is misspelled as "investigettor".

Overall, while the study contributes valuable insights into hookworm re-infection rates and predictors among schoolchildren in northwest Ethiopia, it also advances our understanding of hookworm transmission dynamics and improves public health outcomes in affected communities.

Reviewer #2: The authors performed a study regarding hookworm re-infection among schoolchildren in northwest Ethiopia. This is an interesting study, but it needs some revisions-clarification:

1) Please use the update data and references for lines 38-40.

2) Did you consider nutritional habits, such as eating raw vegetables, etc, for the selected cases.

Reviewer #3: PONE-D-23-41533

Pattern and predictor of hookworm re-infection among schoolchildren in three districts of Amhara Region, northwest Ethiopia

PLOS ONE

Reviewer’s Comments

The manuscript describes the findings of baseline faecal surveys conducted among school children in three districts of Amhara region, northwest Ethiopia, and follow up faecal surveys conducted among hookworm (HW) positive subjects at 4th and 6th month after deworming with albendazole with the objective of determining the reinfection rates of hookworm infection and predictive factors associated with HW reinfection. Although the methodology and results satisfy the study objectives there are many shortcomings in the manuscript preparation compromising the scientific validity of the manuscript. My queries and comments are herewith listed.

1. The Introduction describes the HW infection status as “high prevalence” in rural Amhara region. Is there any information in the literature regarding the species distribution of HWs causing infection in the region? Which may be useful to include in the Introduction.

2. Likewise, the endemicity of other geohelminth infections in the region if known should be included in the introduction.

3. Line 41-42, “despite control measures implemented…..” Please include a brief description of the control measures implemented under Introduction or discussion.

4. Under Methods, a brief description of the areas studied need to be included such as climatic conditions including vegetation, water supply, sanitary facilities, rural or urban (which has been mentioned) and routine deworming programs if any etc.

5. Brief description of sample size calculation to be included under methodology.

6. Under results, the baseline faecal surveys report only HW infections which is surprising in the presence of widespread faecal contamination of the environment as described in this manuscript. Probable explanations need to be included in the discussion as to why there is only a high prevalence of HW infections and absence of other geohelminth species.

7. There were multiple spelling and grammatical errors, some of which I have highlighted in the edited manuscript. The manuscript could be improved with English language editing.

6. PLOS authors have the option to publish the peer review history of their article (what does this mean?). If published, this will include your full peer review and any attached files.

Reviewer #1: No

Reviewer #2: No

Reviewer #3: No

---

## [Author Response · Author response to Decision Letter 0]

16 Apr 2024

Authors ‘ response to reviewers’ comments

First, we would like acknowledge the editor and the reviewers for your constructive comments and questions. We have learned much from your comments and we have tried to respond to your comments in the manuscript. 

PONE-D-23-41533

Pattern and predictor of hookworm re-infection among schoolchildren in three districts of 

Journal Requirements:

Authors’ response: The PLOS ONE style has been maintained. 

2. We note that another paper by the same author group and with the title [“Performance Evaluation of Three Diagnostic Methods for Soil-Transmitted Helminth Infections among Schoolchildren in Amhara Region, Northwest Ethiopia] was recently published. It has been noted that the two submissions may have some overlap with regard to the population cohort. We ask that you please clarify why you feel the two reports should be considered as separate articles in your cover letter. In your manuscript, please ensure you cite, discuss, and acknowledge overlap with the related work, and provide adequate justification (in the Introduction) for the new submission in light of the related published work.

Authors’ response: I appreciate the editors concern. In our project, we had two main objectives. The first objective was to evaluate the performance of diagnostic methods against STHs in the baseline data. The second objective was to identify the patterns and predictor factors of hookworm infection after treatment has been given. Although the two articles are conducted in the same population, the have different objectives. In any case I cited the first title in this manuscript (line 267-68)

Authors’ response: “All relevant data are within the manuscript and its Supporting Information files has been included in the manuscript.”

Authors’ response: Our entire data will need to be made freely accessible if our manuscript is accepted for publication.

Authors’ response: I have already had ORCID iD

Authors’ response: Okay!

Reviewers' comments:

Comments to the Author

1. Is the manuscript technically sound, and do the data support the conclusions?

Reviewer #1: Partly

Reviewer #2: Yes

Reviewer #3: Yes

Authors’ response: The conclusions are drown from the main findings of the study.

2. Has the statistical analysis been performed appropriately and rigorously?

Reviewer #1: Yes

Reviewer #2: I Don't Know

Reviewer #3: I Don't Know

Authors’ response: The main findings have been analyzed properly.

3. Have the authors made all data underlying the findings in their manuscript fully available?

Reviewer #1: No

Reviewer #2: Yes

Reviewer #3: Yes

Authors’ response: All data which supports the findings have been available in the manuscript.

4. Is the manuscript presented in an intelligible fashion and written in standard English?

Reviewer #1: Yes

Reviewer #2: Yes

Reviewer #3: No

Authors’ response: The language is clear and correct and has been improved based on the given comments.

5. Review Comments to the Author

Reviewer #1: The study by Belay et. al., titled "Patterns of Hookworm Reinfection Rates and Predictors among Schoolchildren in Northwest Ethiopia" aims to investigate the re-infection rates and predictors of hookworm among schoolchildren in northwest Ethiopia. While the study addresses an important public health issue, there are a few queries related to its design and methodology to assess the robustness of the findings and the implications for future research and interventions.

Authors’ response: Thank you for comment.

Major Queries:

External Validity: To what extent can the findings of the study be generalized to other populations of schoolchildren in Ethiopia or similar settings with high hookworm prevalence? What measures were taken to ensure the representativeness of the study sample?

Authors’ response: I appreciate the reviewer’s concern. The current study was conducted in three districts. One school was selected in each district. To insure the representativeness of the study, districts, schools and schoolchildren was selected randomly. In addition, to make it representative, two schools were selected from the rural and one from the city. So, the data obtained in this study can be generalized to other populations of schoolchildren in Ethiopia or similar settings with high hookworm prevalence.

Methodological Rigor: How were potential sources of bias, such as selection bias or measurement bias, addressed during the study design and data collection phases? Were steps taken to minimize the risk of misclassification or misreporting of predictor variables?

Authors’ response: To address the selection bias during checking the patterns of re-infection rate, first we selected those schoolchildren who were positive in the baseline data and treated with albendazole and became negative at day 14th. All schoolchildren negative by the three diagnostic methods were selected to avoid selection bias using different diagnostic methods. Similar laboratory testing was also done in the 4th and 6th months. To mimimize the bias, misclassification and misreporting were check at the 14th day of post treatment, 4th and 6th months of checking. 

Longitudinal Dynamics: Given the short follow-up period of six months, how do the observed re-infection rates and predictors align with longer-term patterns of hookworm transmission and control? What implications do these findings have for the sustainability of intervention strategies over time?

Authors’ response: I appreciate the reviewer’s concern regarding the short follow-up period of six months. I understand the long term follow-up overweight’s the short follow-up period. Although our study was followed for six months due to limited budget, the aliment of predictors in the 4th and 6th month was interesting. In hookworm endemic areas, and people are practicing bare footed, the re-infection rate occurs in very short period. Therefore, the intervention strategy should be strongly advocated using integrated approach (WASH, MDA and Health education). Despite the study conducted in short period of time, it advances the understanding of hookworm transmission dynamics and improves public health outcomes in affected communities, especially in endemic areas.

Comprehensive Assessment: Are there additional predictor variables or potential confounders that were not included in the analysis but may significantly influence hookworm re-infection rates among schoolchildren? How could future research efforts broaden the scope of predictor assessment to capture a more comprehensive range of factors?

Authors’ response: I appreciate the reviewers concern regarding additional variables. There are some environmental factors such as soil type, temperature, vegetation, rain fall and wet environment which might have an influence on hookworm re-infection rates. These factors were not assessed in this study due to limited budget. Further comprehensive assessment of predictor variable including personal, habits, and environmental factors should be conducted to address the re-infection rates of hookworm re-infection rates. 

Intervention Strategies: Based on the identified predictors of hookworm re-infection, what targeted intervention strategies or public health interventions could be implemented to reduce transmission rates and improve outcomes among schoolchildren in northwest Ethiopia? How can these strategies be tailored to address the specific needs and challenges of the local community? Some discussion about them should be included in the manuscript.

Authors’ response: I appreciate the reviewers regarding targeted interventions. To reduce transmission rates and decrease the outcomes of hookworm re-infection rate among schoolchildren, emphasis should be given for prevention using an integrated approach, which includes proper implementation of water sanitation and hygiene, especially in the rural settings, periodic deworming using albendazole and community based health education (how hookworm infection is transmitted and prevented?) should be conducted in northwest Ethiopia. Based on our results, proper latrine utilization and shoes wearing during outdoor activities like farming and irrigation should be targeted to intervene hookworm re-infection. 

Minor point: Line 124 -the word principal investigator is misspelled as "investigettor".

Authors’ response: Correction has been done (line 131).

Overall, while the study contributes valuable insights into hookworm re-infection rates and predictors among schoolchildren in northwest Ethiopia, it also advances our understanding of hookworm transmission dynamics and improves public health outcomes in affected communities.

Authors’ response: Thank you very much!

Reviewer #2: The authors performed a study regarding hookworm re-infection among schoolchildren in northwest Ethiopia. This is an interesting study, but it needs some revisions-clarification:

1) Please use the update data and references for lines 38-40.

Authors’ response: Updated reference have been used (line 38-40).

2) Did you consider nutritional habits, such as eating raw vegetables, etc, for the selected cases.

Authors’ response: Since faeco-oral transmission of hookworm is rare, we could not considered nutritional habits.

Reviewer#3: PONE-D-23-41533

Pattern and predictor of hookworm re-infection among schoolchildren in three districts of Amhara Region, northwest Ethiopia

PLOS ONE

Reviewer’s Comments

The manuscript describes the findings of baseline faecal surveys conducted among school children in three districts of Amhara region, northwest Ethiopia, and follow up faecal surveys conducted among hookworm (HW) positive subjects at 4th and 6th month after deworming with albendazole with the objective of determining the reinfection rates of hookworm infection and predictive factors associated with HW reinfection. Although the methodology and results satisfy the study objectives there are many shortcomings in the manuscript preparation compromising the scientific validity of the manuscript. My queries and comments are herewith listed.

1. The Introduction describes the HW infection status as “high prevalence” in rural Amhara region. Is there any information in the literature regarding the species distribution of HWs causing infection in the region? Which may be useful to include in the Introduction.

Authors’ response: As far as my knowledge is concerned, there is no previous hookworm species data in the region.

2. Likewise, the endemicity of other geohelminth infections in the region if known should be included in the introduction.

Authors’ response: The endemicity of other geohelminth 

---

## [Decision Letter · Decision Letter 1]

26 Apr 2024

Pattern and predictor of hookworm re-infection among schoolchildren in three districts of Amhara Region, northwest Ethiopia

PONE-D-23-41533R1

Dear Dr.Tadesse Hailu Jember

We’re pleased to inform you that your manuscript has been judged scientifically suitable for publication and will be formally accepted for publication once it meets all outstanding technical requirements.

Within one week, you’ll receive an e-mail detailing the required amendments. When these have been addressed, you’ll receive a formal acceptance letter, and your manuscript will be scheduled for publication.

Kind regards,

Tsegaye Alemayehu, MSc

Academic Editor

PLOS ONE

Additional Editor Comments (optional):

Reviewers' comments:

Reviewer's Responses to Questions

**Comments to the Author**

1. If the authors have adequately addressed your comments raised in a previous round of review and you feel that this manuscript is now acceptable for publication, you may indicate that here to bypass the “Comments to the Author” section, enter your conflict-of-interest statement in the “Confidential to Editor” section, and submit your "Accept" recommendation.

Reviewer #2: All comments have been addressed.

2. Is the manuscript technically sound, and do the data support the conclusions?

Reviewer #2: Yes

3. Has the statistical analysis been performed appropriately and rigorously? 

Reviewer #2: I Don't Know

4. Have the authors made all the data underlying the findings in their manuscript fully available?

The PLOS Data policy requires authors to make all data underlying the findings described in their manuscript fully available without restriction, with rare exception (please refer to the Data Availability Statement in the manuscript PDF file). The data should be provided as part of the manuscript or its supporting information or deposited to a public repository. For example, in addition to summary statistics, the data points behind means, medians and variance measures should be available. If there are restrictions on publicly sharing data—e.g. participant privacy or use of data from a third party—those must be specified.

Reviewer #2: Yes

5. Is the manuscript presented in an intelligible fashion and written in standard English?

Reviewer #2: Yes

6. Review Comments to the Author

Reviewer #2: Thank you for your responses and revisions to the manuscript. The manuscript is suitable for publication.

7. PLOS authors have the option to publish the peer review history of their article (what does this mean?). If published, this will include your full peer review and any attached files.

If you choose “no”, your identity will remain anonymous, but your review may still be made public.

Reviewer #2: No

---

## [Editor Report · Acceptance letter]

15 May 2024

PONE-D-23-41533R1 

PLOS ONE

Dear Dr. Hailu, 

I'm pleased to inform you that your manuscript has been deemed suitable for publication in PLOS ONE. Congratulations! Your manuscript is now being handed over to our production team.

Kind regards, 

on behalf of

Dr. Tsegaye Alemeyhu 

Academic Editor

PLOS ONE